# Ego-Foresight: Self-supervised Learning of Agent-Aware Representations for Improved RL

**Manuel Serra Nunes**[1*]**, Atabak Dehban**[1]**, Yiannis Demiris**[2]**, José Santos-Victor**[1]
[1] Institute for Systems and Robotics, Instituto Superior Técnico, U. Lisboa
[2] Personal Robotics Laboratory, Imperial College London

## Abstract

Despite the significant advances in Deep Reinforcement Learning (RL) observed in the last decade, the amount of training experience necessary to learn effective policies remains one of the primary concerns in both simulated and real environments. Looking to solve this issue, previous work has shown that improved efficiency can be achieved by separately modeling the agent and environment, but usually requires a supervisory signal. In contrast to RL, humans can perfect a new skill from a small number of trials and often do so without a supervisory signal, making neuroscientific studies of human development a valuable source of inspiration for RL. In particular, we explore the idea of motor prediction, which states that humans develop an internal model of themselves and of the consequences that their motor commands have on the immediate sensory inputs. Our insight is that the movement of the agent provides a cue that allows the duality between the agent and environment to be learned. To instantiate this idea, we present Ego-Foresight (EF), a self-supervised method for disentangling agent information based on motion and prediction. Our main finding is that, when used as an auxiliary task in feature learning, self-supervised agent-awareness improves the sample-efficiency and performance of the underlying RL algorithm. To test our approach, we study the ability of EF to predict agent movement and disentangle agent information. Then, we integrate EF with model-free and model-based RL algorithms to solve simulated control tasks, showing improved sample-efficiency and performance.

## 1 Introduction

While it usually goes unnoticed as we go about our daily lives, the human brain is constantly engaged in predicting imminent future sensori inputs (Clark, 2016). This happens when we react to a friend extending their arm for a handshake, when we notice a missing note in our favorite song, and when we perceive movement in optical illusions (Watanabe et al., 2018). At a more fundamental level, this process of predicting our sensations is seen by some as the driving force behind perception, action, and learning (Friston, 2010). But while predicting external phenomena is a daunting task for a brain, motor prediction (Wolpert & Flanagan, 2001) - i.e., predicting the sensori consequences of one's own movement - is remarkably simpler, yet crucial in increasing saliency of external events (Blakemore et al., 1999). Anyone who has ever tried to self-tickle has experienced the brain in its predictive endeavors: in trying to predict external (and more critical) inputs, the brain is thought to suppress self-generated sensations, to increase the saliency of those coming from outside - making it hard to feel self-tickling (Blakemore et al., 2000).

In artificial systems, a significant effort has been devoted to learning World Models (Ha & Schmidhuber, 2018; Finn et al., 2016; Gumbsch et al., 2024), which are designed to predict future states of the whole environment and allow planning in the latent space. Despite encouraging deployments of these models in real-world robotic learning (Wu et al., 2023), their application remains restricted to safe and simplified workspaces, with sample-efficient Deep Reinforcement Learning (RL) being one of the main challenges.

---

*Corresponding author. Email to manuelserranunes@tecnico.ulisboa.pt

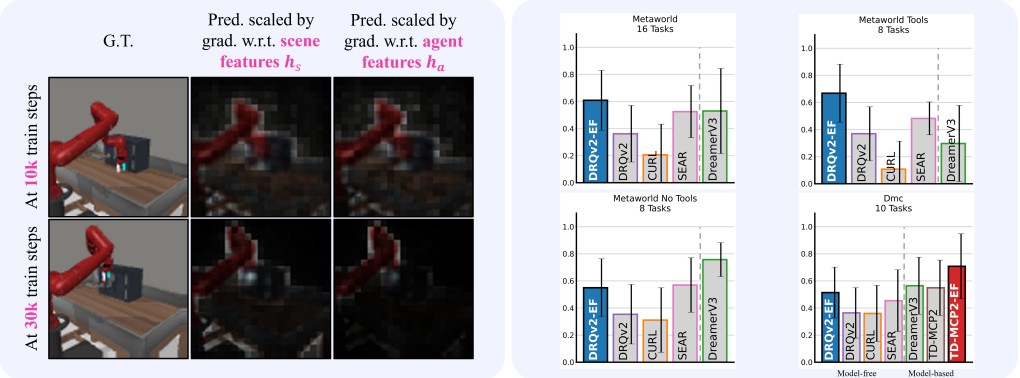

Figure 1: (left) Reconstructed image scaled by the intensity of the gradient with respect to different sections of the learned feature vector, showing the emergence of agent-aware feature representations. (right) In most RL tasks, an algorithm's Efficiency Normalized Score (see 4.3) improves when augmented with Ego-Foresight, sometimes outperforming supervised and model-based methods. The dashed line separates model-free from model-based algorithms.

Although comparatively less explored, the idea of separately modeling the agent and the environment has also been investigated in RL, with previous work demonstrating improved sample-efficiency in simulated control tasks (Gmelin et al., 2023). Additionally, this type of approach has been used to allow zero-shot policy transfer between different robots (Hu et al., 2022) and to improve environment exploration (Mendonca et al., 2023). Common to all these works is the reliance on supervision to obtain information about the appearance of the robot, allowing explicit disentangling of the agent from the environment. This is usually provided in the form of a mask of the agent within the scene, which is obtained either from geometric IDs in simulation, by fine-tuning a segmentation model, or even by resorting to the CAD model of the robot. In a real-world robotic setting, it usually means the addition of a separate system to segment the robot, adding complexity to the setup. Furthermore, supervised approaches are tied to the body-schema of the agent and cannot adapt when it changes, for example, when using tools.

As humans, we do not receive such a detailed and hard-to-obtain supervisory signal and yet, during our development, we build a representation of ourselves (Watson, 1966) capable of adapting both slowly, as we grow, and fast, when we pick up tools (Maravita & Iriki, 2004). In this work, we argue that unsupervised awareness of *self* can also be achieved in artificial systems, and study its advantages relative to supervised methods.

In our approach, which we name Ego-Foresight (EF), we place the agent's embodiment as an intrinsic part of the learning process, since it determines the visuomotor sensations that the agent can expect as it moves. Our insight is that agent information can be disentangled by having the agent move, while trying to predict the visual changes to its body configuration (Fig.1 (left)), and that awareness and prediction of the agent's movements should improve its ability to solve complex tasks (Fig.1 (right)).

In our implementation, we use a convolutional encoder that receives as input a limited amount of context RGB frames. A recurrent model takes the visual features and sequence of planned actions and predicts the future configurations of the agent, which the decoder reconstructs to obtain the future frames. This framework naturally lends itself to self-supervised training. In experiments with simulated data, we demonstrate the ability to predict the movement of the robot arm while ignoring moving objects in the environment and the integration of tools as part of the body-schema.

Furthermore, we extend existing model-free and model-based RL algorithms with Ego-Foresight, and assess the modified algorithms in two simulated domains and on different control tasks involving a diverse range of embodiments, demonstrating that our approach can improve sample-efficiency and performance. We consider that two main factors contribute to this result: i) the disentanglement of agent information allows the RL algorithm to focus its capacity on learning the control of the agent in the initial stages of training, and later on the external aspects and potential interactions within the

environment, and ii) imposing predictability in the robot's movements regularizes the RL algorithm. In summary, the key contributions of this paper are the following:

- We propose a self-supervised method for disentangling agent information based on motion and self-prediction, removing the supervision required by previous methods.
- We demonstrate the ability of our method to reconstruct future configurations of an agent and adapt to changes in its body-schema.
- We extend two state-of-the-art RL algorithms (DrQ-v2 and TD-MPC2) with Ego-Foresight, showing improvement in sample-efficiency and performance in simulated control tasks.[1]
- We conduct an ablation study on the hyperparameters introduced by our approach.

## 2 BACKGROUND

**Model-free visual RL**  RL problems are typically formulated as a Markov Decision Problem, defined as a tuple $(\mathcal{S}, \mathcal{A}, \mathcal{T}, \mathcal{R}, \gamma, d_0)$, where $\mathcal{S}$ is the state space, $\mathcal{A}$ is the action space, $\mathcal{T}(\boldsymbol{s}_{t+1}|\boldsymbol{s}_t, \boldsymbol{a}_t)$ is the transition function, $\mathcal{R}(\boldsymbol{s}_t, \boldsymbol{a}_t)$ is the reward function, $\gamma \in [0, 1]$ is a discount factor and $d_0$ is the distribution over the initial states $\boldsymbol{s}_0$. The objective in RL is to learn the policy $\pi : \mathcal{S} \to \mathcal{A}$ that maximizes the expected discounted cumulative reward $\mathbb{E}_\pi[\sum_{t=0}^{\infty} \gamma^t \mathcal{R}(\boldsymbol{s}_t, \boldsymbol{a}_t)]$, with $\boldsymbol{a}_t \sim \pi(\cdot|\boldsymbol{s}_t)$ and $\boldsymbol{s}_{t+1} \sim \mathcal{T}(\cdot|\boldsymbol{s}_t, \boldsymbol{a}_t)$.

Over the last decade, work in RL from images (Mnih et al., 2013), where environment representations are learned from high-dimensional inputs, has allowed agents to solve problems for which features cannot be designed by experts. A widely used algorithm for RL from pixels is DDPG (Lillicrap et al., 2016), an off-policy actor-critic algorithm for continuous control, which alternates between learning an approximator to the Q-function $Q_\phi$ and a deterministic policy $\pi_\theta$. DrQ-v2 (Yarats et al., 2021) introduces an augmentation technique for the input images, extending DDPG (Lillicrap et al., 2016) for improved efficiency. Additionally, it adopts the Twin Delayed variation of DDPG (Fujimoto et al., 2018), which adds clipped double Q-learning and delayed policy updates to limit overestimation of the Q-values. Having sampled a batch of transitions $\tau = (\boldsymbol{s}_t, \boldsymbol{a}_t, r_{t:t+n-1}, \boldsymbol{s}_{t+n})$ from the replay buffer $\mathcal{D}$, $Q_\phi$ is learned by minimizing the mean-squared Bellman error:

$$\mathcal{L}_{critic}(\phi, \mathcal{D}) \doteq \mathbb{E}_{\tau \sim \mathcal{D}} \left[ (Q_{\phi_k}(\boldsymbol{s}_t, \boldsymbol{a}_t) - y)^2 \right] \quad \forall k \in \{1, 2\}, \tag{1}$$

using target networks $Q_{\hat{\phi}_k}$ to approximate the target values, with $n$-step returns, and where $\hat{\phi}_k$ are slowly updated copies of the parameters $\phi_k$ :

$$y \doteq \sum_{i=0}^{n-1} \gamma^i r_{t+i} + \gamma^n \min_{k=1,2} Q_{\hat{\phi}_k}(\boldsymbol{s}_{t+n}, \pi_\theta(\boldsymbol{s}_{t+n})). \tag{2}$$

The policy is learned by maximizing $\mathbb{E}_{s \sim D}[Q_\phi(s_t, \pi_\theta(s_t))]$, to find the action that maximizes the Q-function. Improving sample-efficiency in RL from pixels remains an open problem, and new methods for mitigating it have been consistently proposed (Hessel et al., 2018; Laskin et al., 2020; Stooke et al., 2021; Zheng et al., 2023).

**Model-based visual RL**  In model-based RL, approaches such as Dreamer-v3 (Hafner et al., 2025) and TD-MPC2 (Hansen et al., 2024) have achieved remarkable results by exploring the idea of training in an imagined latent space where thousands of parallel trajectories can be sampled with a learned world model. In this work, we modify TD-MPC2 to include Ego-Foresight as an auxiliary loss term. The architecture of TD-MPC2 consists of five components:

$$\begin{array}{llll} \text{Encoder:} & \boldsymbol{h}_t = E(\boldsymbol{s}_t) & \text{Policy prior:} \; \hat{\boldsymbol{a}}_t = \pi(\boldsymbol{h}_t) & \text{Rew.:} \; \hat{r}_t = R(\boldsymbol{h}_t, \boldsymbol{a}_t) \\ \text{Dynamics:} & \boldsymbol{h}_{t+1} = d(\boldsymbol{h}_t, \boldsymbol{a}_t) & \text{Term. value:} \; \hat{q}_t = Q(\boldsymbol{h}_t, \boldsymbol{a}_t) \end{array} \tag{3}$$

Model updates are done in two steps, with world-model updates using data from interactions stored in the replay buffer and optimizing the loss:

$$\mathcal{L}_{TDMPC}(\theta, \mathcal{D}) \doteq \mathbb{E}_{\tau_{0:H} \sim \mathcal{D}} \left[ \sum_{t=0}^{H} \lambda^t (\|\boldsymbol{h}_{t+1} - \text{sg}(E(\boldsymbol{s}_{t+1}))\|_2^2 + \text{CE}(\hat{r}_t, r_t) + \text{CE}(\hat{q}_t, q_t)) \right], \tag{4}$$

---

[1]Media available at: https://mserranunes.github.io/ego-foresight

where $\theta$ are the parameters of $E$, $d$, $R$ and $Q$, $\lambda$ is a temporally dependent discount factor, $H$ is the MPC horizon, sg is the *stop-gradient* operator, CE is the cross-entropy loss, and $q_t$ is the TD-target. The policy prior $\pi$ is used to warm-start the planning procedure and accelerate its convergence and is optimized with maximum entropy. For planning, TD-MPC2 uses Model Predictive Path Integral by taking advantage of the dynamics model $d$ to rollout the latent trajectories that result from sampled action sequences and obtaining an estimate of their return using $Q$ and $R$.

# 3 APPROACH

## 3.1 EPISODE PARTITION

We define our dataset as a collection of $N$ episodes $\boldsymbol{e}_i$ of fixed length $L$ (Fig. 2), which include a sequence of observations in the form of RGB frames $\boldsymbol{x}$ and the corresponding actions $\boldsymbol{a}$ of the agent: $\boldsymbol{e}_i = \{(\boldsymbol{x}_0, \boldsymbol{a}_0), ..., (\boldsymbol{x}_L, \boldsymbol{a}_L)\}_i$, $i = 1, ..., N$. During training, we randomly select a window of size $C + H$ within each episode - which corresponds to the number of context frames plus the prediction horizon. This artificially augments the available data by creating different observations within each episode. Finally, from each window, $C$ context steps are taken for input to the model, as well as the actions for the whole sequence up to $t_{C+H}$. The frames between $t_C$ and $t_{C+H}$ are used as target for the prediction.

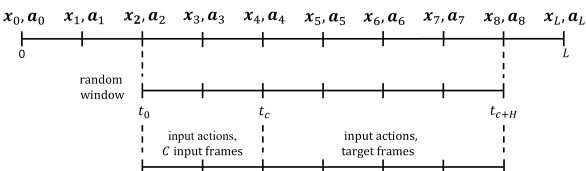

Figure 2: Partition of an episode $\boldsymbol{e}_i$. From each sequence of frames and actions (top), a window of size $C + H$ is randomly sampled (middle). The first $C$ steps of the window are used as context. For the remaining steps, the actions are used as input, while the frames serve as the target for the prediction.

## 3.2 MOTION-BASED AGENT DISENTANGLEMENT

To model future visual configurations of the agent, we propose an encoder-decoder model (Fig. 3) with a recurrent block that predicts the next agent configuration in feature space. The encoder, parameterized by $\psi$, produces a feature representation $\boldsymbol{h} \in \mathbb{R}^n$, obtained from the context frames, $\boldsymbol{h}^{t_c} = E_\psi(\boldsymbol{x}^{t_0:t_c})$. This vector is then split into scene $\boldsymbol{h}_s^{t_c} \in \mathbb{R}^m$ and agent $\boldsymbol{h}_a^{t_c} \in \mathbb{R}^l$ features, with $l + m = n$. The agent features are used as input to the recurrent block, along with the action at the next time step $\boldsymbol{a}^{t_c+1}$, for prediction of the next agent configuration. The recurrent network keeps predicting agent features $\hat{\boldsymbol{h}}_a^{t_j+1} = FC_\psi\left(\hat{\boldsymbol{h}}_a^{t_j}, \boldsymbol{a}^{t_j+1}\right)$ until a time step $t_k$, randomly sampled inside the prediction horizon in each training update. Finally, $\hat{\boldsymbol{h}}_a^{t_k}$ is concatenated with $\boldsymbol{h}_s^{t_c}$ and passed to the decoder for reconstruction of $\hat{\boldsymbol{x}}^{t_k} = D_\psi\left(\boldsymbol{h}_s^{t_c}, \hat{\boldsymbol{h}}_a^{t_k}\right)$. The components of the feature space corresponding to $\boldsymbol{h}_s^{t_c}$ hold information about the visual content of the whole scene, which isn't present in $\hat{\boldsymbol{h}}_a^{t_k}$ but is necessary for reconstruction. The components corresponding to $\hat{\boldsymbol{h}}_a^{t_k}$ are then responsible for providing information to the decoder about the future configuration of the agent. Because scene information is fast-forwarded from $t_c$, the reconstructed frames tend to be close to $\boldsymbol{x}^{t_c}$ except for the configuration of the agent for which there is a prediction at $t_k$. This formulation results in the following reconstruction loss term:

$$\mathcal{L}_{ef}(\psi, \mathcal{D}) = \mathbb{E}_{\tau \sim \mathcal{D}}\left[||\hat{\boldsymbol{x}}^{t_k} - \boldsymbol{x}^{t_k}||_2^2\right]. \tag{5}$$

Crucially, we set the dimensionality $l$ of $\boldsymbol{h}_a$ to be a small fraction of $n$, to create a bottleneck that leads the recurrent model to focus its capacity on the most predictable dynamics of the environment, which are those of the agent itself. Furthermore, by fast-forwarding the scene's content features $\boldsymbol{h}_s$ encoded from the context frames to the reconstruction time step $t_k$, the recurrent block is discouraged from predicting the dynamics of the complete environment, forcing the agent features to be concentrated on $\boldsymbol{h}_a$.

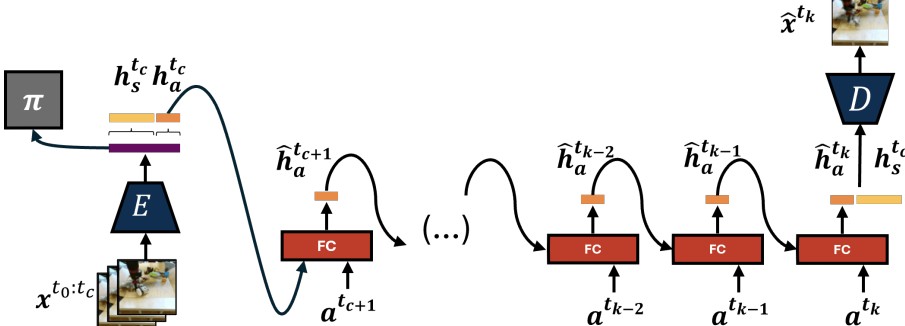

Figure 3: Visuomotor prediction using Ego-Foresight.

## 3.3 AGENT VISUOMOTOR PREDICTION AS FEATURE LEARNING FOR RL

To test how our method affects sample-efficiency in RL, we implement it on top of two of the best performing RL algorithms for visual control: DrQ-v2 (Yarats et al., 2022), an off-policy model-free RL algorithm, and TD-MPC2, a model-based algorithm. We name the extended algorithms DrQv2-EF and TD-MPC2-EF. The episodes stored in the replay buffer $\mathcal{D}$ allow us to maintain the approach described in section 3.2 while jointly learning the policy. Despite our option for these algorithms, we preserve the generality of the approach so that it can be applied to any model-free or model-based algorithm that makes use of experience replay and a visual encoder.

To extend an RL algorithm with Ego-Foresight, we take the feature vector at the output of the visual encoder and induce the disentanglement of agent information using the approach described in Section 3.2. Whereas the encoder is shared with the baseline algorithm, the recurrent unit and the decoder represent additional modules. An optimization step is then performed by augmenting the loss function of the baseline algorithm with $\mathcal{L}_{ef}$ (5), with the losses backpropagating to the encoder.

Although this approach will fit most RL algorithms using a visual encoder, some may require more nuanced integration, for example, actor-critic algorithms in which the optimization is done in two steps. In the case of DrQ-v2 and TD-MPC2, the policy neural network is separate from the feature learning process, simply receiving the low-dimensional feature vector $h_{t_c}$ coming from the encoder in the forward pass, while its gradients do not flow to the encoder in the backward pass. As such, in DrQ-v2 we optimize $\mathcal{L}_{ef}$ together with the critic loss of equation 1, resulting in a final objective function that is the weighted sum of both terms:

$$\mathcal{L}(\phi, \psi, \mathcal{D}) \doteq \mathcal{L}_{critic}(\phi, \mathcal{D}) + \beta \mathcal{L}_{ef}(\psi, \mathcal{D}). \tag{6}$$

In the original DrQ-v2 objective, the visual encoder is optimized via gradients coming from the critic network, and therefore it learns to extract features that are predictive of value. The addition of $\mathcal{L}_{ef}$ can be viewed as a regularizer on the learned features, as it imposes them to be predictive of the agent's motion and is independent from the reward and the task.

In line with the model-free case, for TD-MPC2 we introduce $\mathcal{L}_{ef}$ as an additional loss term in the model objective of equation 4, jointly optimizing

$$\mathcal{L}(\theta, \psi, \mathcal{D}) \doteq \mathcal{L}_{TDMPC}(\theta, \mathcal{D}) + \beta \mathcal{L}_{ef}(\psi, \mathcal{D}). \tag{7}$$

When training Ego-Foresight together with an RL algorithm, we notice that the agent tends to perform goal-directed movements as it seeks to maximize reward, preventing the observation of diverse enough motions to learn the visuomotor mapping. To solve this issue we introduce an optional motor-babbling (Saegusa et al., 2009; Kase et al., 2021) stage for a fixed number of steps at the start of training, during which actions are random choices of $\pm 1$, forcing exploratory movements. This means that the predictive ability for agent features is developed mainly during the initial stages of training. After the babbling stage, Ego-Foresight continues to be optimized according to equation 6, allowing adaptation in later stages, namely when the agent begins to pick-up an available tool.

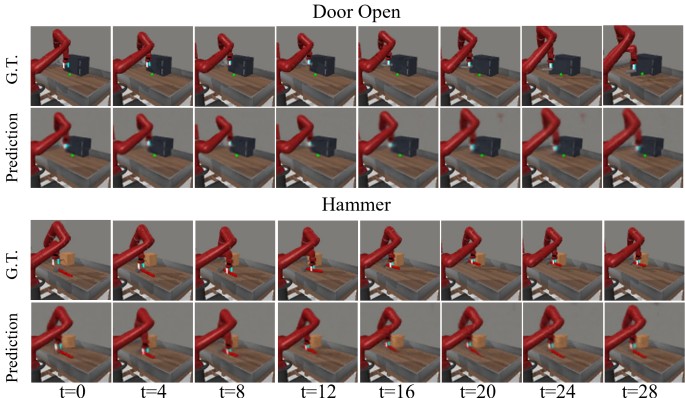

Figure 4: The policy's solution to two tasks and the model prediction for the sequence of actions that solves the task. Notably, the hammer is integrated into the agent and therefore predicted.

## 4 EXPERIMENTS & RESULTS

### 4.1 EXPERIMENTAL SETUP

We conduct our experiments in simulation, with tests on two challenging RL benchmarks: Meta-world (Yu et al., 2020) and DMC (Tassa et al., 2018). The chosen benchmarks are implemented in MuJoCo (Todorov et al., 2012) and provide a wide variety of tasks, ranging from robotic object manipulation to locomotion, and include a diverse set of embodiments. In all environments, we provide camera observations and the sequence of commanded actions as inputs to the model. In DMC we choose the 10 tasks for which official TD-MPC2 results from pixels are available online. For Meta-World, the tasks were chosen so that half would require the use of tools and the other half would require interaction with different kinds of objects.

### 4.2 AGENT-ENVIRONMENT DISENTANGLEMENT

The experiments in this section intend to answer two fundamental questions in our study: **(1)** Is Ego-Foresight able to disentangle agent information and make predictions consistent with the ground truth? **(2)** Is our approach able to adapt to changing embodiments?

**Visualizing encoded features**    To study whether the proposed method for disentangling learns to concentrate information related to the agent in the feature vector $h_a$, we start by visualizing the areas of the reconstructed frames that are influenced the most by changes in the components of $h_a$. To do this, we use DrQv2-EF and split the reconstructed frame into patches of $4 \times 4$ pixels and compute the gradient of each patch with respect to the section of interest of the feature vector at different moments during a training run. We then scale the intensity of each patch proportionally to the gradient. In Fig. 1 (left) it is possible to observe that at the start of training (after 10k steps), both $h_a$ and $h_s$ have a similar influence in the predicted frame, with solid and static regions such as the background being quickly overfitted by the decoder. As training progresses, $h_s$ continues to encode all the varying visual aspects of the scene, including the cabinet and the table borders (which change due to image augmentation), while $h_a$ becomes specialized in agent information. We provide additional visualizations in Appendix A.5.

In Fig. 4, we highlight two tasks using DrQv2-EF, showing the sequence of actions taken by the agent to solve them and how, for those action sequences, the motion of the agent is correctly predicted by EF. In the Door Open task, the robot starts by moving the gripper away from the camera toward the handle of the door and, after making contact with it, pulls back while opening the door. We highlight that in the predicted frames the door is reconstructed using visual content information from $h_s^{t_c}$, and therefore remains mostly static, as part of the scene. This contrasts with the Hammer task, where in addition to the movement of the agent, the model also predicts how the

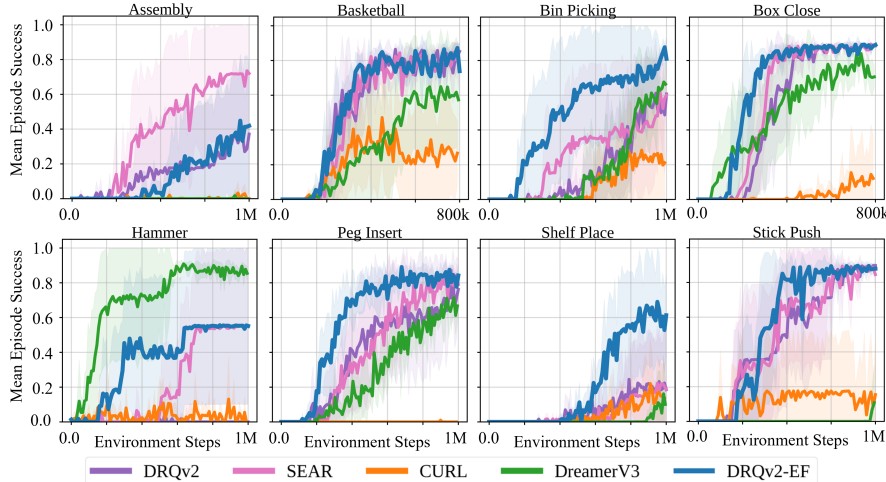

Figure 5: Per-task results on Meta-World tasks that require tool use.

hammer moves. This is due to the fact that once the hammer is picked up, its motion is completely determined by the movement of the robot - effectively being integrated into the body of the robot - and can therefore be predicted from the future actions. The same is not true in the Door Open task, where there is no consistency between the movement of the door and that of the agent from episode to episode. This highlights the ability of our approach to adapt to changing embodiments, something that would not be possible in a supervised setting. We note that in both visualizations, EF was trained with an horizon of 10 frames and used for prediction until time step 28, resulting in some divergence from the ground-truth motion and in blurry predictions.

## 4.3 AGENT-AWARE REPRESENTATIONS FOR IMPROVED REINFORCEMENT LEARNING

In this section, we focus on the effect of Ego-Foresight on RL control tasks, intending to answer two central questions: **(1)** Can EF improve the sample-efficiency and performance of RL algorithms, in particular in tasks requiring tools? **(2)** How do the design choices affect the success of RL tasks?

**Comparison Axes** We present both per-task results and aggregate results across benchmarks and focus on sample-efficiency, measured in the number of environment steps and on performance in terms of success rate or reward, depending on the benchmark. For each task, we perform 5 runs per baseline using different random seeds and report mean and standard deviation (for TD-MPC2 on DMC we use the 3 seeds available in the official code repository). To measure sample-efficiency and performance together, we define an Efficiency Normalized Score (ENS), with results presented in Fig. 1 (right). To compute ENS, we find the step at which 95% of the maximum performance achieved by any baseline on a given task is reached. Then, we measure the performance of each algorithm at that point. For robustness, this procedure is repeated for the 90% and 85% thresholds, with values being averaged across both thresholds and tasks to obtain the final aggregate score.

**Baselines** With EF being designed to serve as an auxiliary task for improved feature learning in existing RL methods, the main baseline for an extended algorithm is the original algorithm itself. As such, we focus mainly on the comparison of DrQv2-EF with DrQ-v2 and TD-MPC2-EF with TD-MPC2. We provide three additional baselines: SEAR (Gmelin et al., 2023), CURL (Laskin et al., 2020) and Dreamer-v3 (Hafner et al., 2025), introduced in Section 2. SEAR builds on top of DrQ-v2, by splitting the feature representation into agent and full environment information, which is achieved with a supervisory mask. SEAR is therefore a close supervised baseline to our approach. In the experiments below, the supervisory mask is obtained directly from the simulator. CURL uses a contrastive objective for self-supervised visual feature extraction. Dreamer-v3 is a model-based RL algorithm that has achieved state-of-the-art results in a wide range of RL tasks. All results are

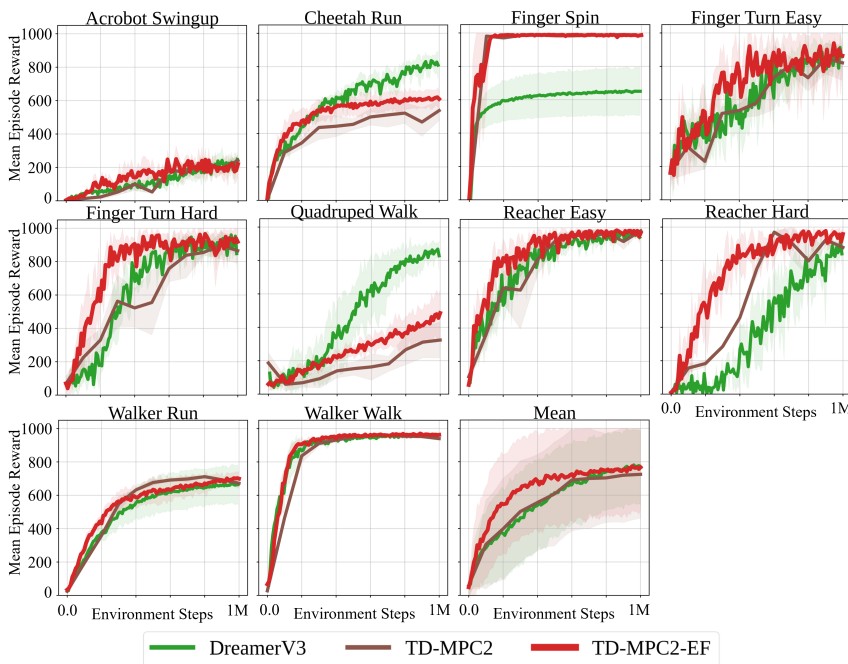

Figure 6: Per-task results on DMC for the model-based RL algorithms.

obtained using the code provided by the authors of each paper. Unless otherwise noticed, we use the default hyperparameters of each baseline, including for DrQv2-EF and TD-MPC2-EF.

**Reinforcement Learning Results** In Figures 1, 5, 6 and in Appendix A.3 we present the aggregate and per-task results on 26 RL tasks. Additionally, we report *Rliable* metrics (Agarwal et al., 2021) in Fig. 7 for a rigorous evaluation of the statistical uncertainty of each baseline. We observe that in 21 of those tasks, augmenting DrQ-v2 with EF leads to improvements over the original algorithm, often reducing the amount of steps necessary to solve the task by a significant margin and in many cases improving the asymptotic performance. Importantly, augmenting DrQ-v2 with EF does not degrade the results in any of the tasks. DrQv2-EF also shows improvement in the ENS and Rliable metrics over the supervised baseline SEAR and over CURL in both Meta-World and DMC and is competitive with Dreamer-v3, one of the best performing models in the literature. Still, in some tasks, these baselines outperform DrQv2-EF, as can be observed in Fig. 5. We attribute this to the diversity of reward functions of the benchmark, which means that some algorithms will adapt better to some tasks than to others. Interestingly, in tasks that require the use of tools (Fig. 5), we notice that the overall performance gap to the baselines increases, when compared with tasks that do not require the use of tools (see Fig. 1 and Appendix A.3). We interpret this result as a consequence of the tools being integrated in the feature representation of the agent, as their movement becomes predictable from the robot's actions when the agent is holding them.

Similar improvements are obtained for TD-MPC2-EF, which improves on 8 out of 10 tasks versus the original TD-MPC2, while obtaining similar results to the original algorithm in the other 2 tasks. In terms of ENS (Fig. 1), a significant improvement is achieved with the addition of our approach. Moreover, the results of TD-MPC2-EF indicate that our approach is not specific to a single architecture or to model-free RL, and that it can benefit other algorithms in the visual RL community.

**Ablation Study** To better understand the effect of the different design choices in this work, we ablate the main hyperparameters introduced by Ego-Foresight on DrQv2-EF and present the results in Fig. 8. By varying the weight $\beta$ of the EF loss term, it is possible to observe how the proposed auxiliary loss has a regularizing effect on learning, improving performance without explicitly optimizing for reward. We also observe that the horizon length does not have a consistent impact on the results and that even for short horizons results improve on DrQ-v2 (see Fig. 11). This indicates

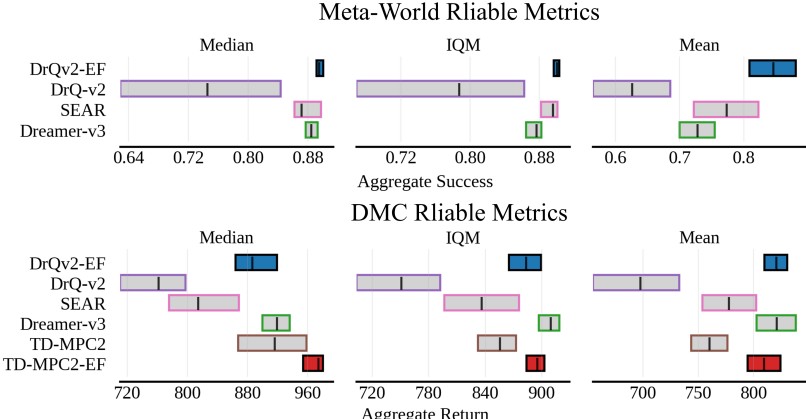

Figure 7: *Rliable* metrics on the 16 Meta-World and 10 DMC tasks with $95\%$ stratified bootstrap confidence intervals using the best evaluation score.

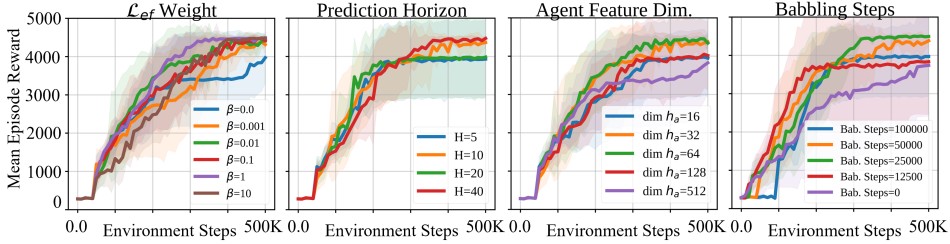

Figure 8: Ablation of the hyperparameters introduced in DrQv2-EF on Meta-World Door Open.

that the model retains the ability of learning the visuomotor mapping for smaller $H$. Nevertheless, horizon values of 10 and 40 show stronger performance, with $H = 10$ being the preferred choice for default value due to lower computational cost. In terms of the dimensionality of the agent features $\boldsymbol{h}_a$, the ablation study shows that larger sizes have a detrimental effect, possibly because a softer bottleneck leads the recurrent block to try to predict other environment features, reducing the ability to disentangle agent information. Additionally, because the dimensionality of $\boldsymbol{h}$ is fixed at $n$ and distributed between $\boldsymbol{h}_s$ and $\boldsymbol{h}_a$, going over the dimension size necessary to represent the state-space of the agent adds no benefit to agent representation while removing representational capacity from $\boldsymbol{h}_s$. Similarly, the introduction of babbling is a meaningful contribution to the results, but when applied for too many steps it can delay learning.

## 5 RELATED WORK

The notion of distinguishing self-generated sensations from those caused by external factors has been studied and referred to under different terms and motivated by a broad range of applications. Originating in psychology and neuroscience, with the study of contingency awareness (Watson, 1966) and sensorimotor learning (motor prediction) (Wolpert et al., 2011; Wolpert & Flanagan, 2001), recently this concept has seen growing interest in AI as an auxiliary learning mechanism.

In developmental robotics, Zhang & Nagai (2018) have focused on this problem from the standpoint of self-other distinction, by employing 8 NAO robots observing each other executing a set of motion primitives, and trying to differentiate *self* from the learned representations. Lanillos et al. (2020) note that to answer the question "Is this my body?", an agent should first learn to answer "Am I generating those effects in the world?". Their robot learns the expected changes in the visual field as it moves in front of a mirror or in front of a twin robot and classifies whether it is looking at itself or not. Our approach is somewhat analogous to these works, in the sense that we identify as being part of the agent what can be visually predicted from the future actions while the robot moves.

Still in robotics, the idea of modeling the agent has connections with work in body perception and visual imagination for goal-driven behaviour (Sancaktar et al., 2020). Another application that has been explored is the learning of modular dynamic models, that decouple robot dynamics from world dynamics, allowing the latter to be reused between robots with different morphologies. Hu et al. (2022) propose a method for zero-shot policy transfer between robots, by taking advantage of robot-specific information - such as the CAD model - to obtain a robot mask from which future states can be predicted, given its dynamics. These future states are then used to solve manipulation tasks using model-based RL. Finally, this concept has been used to ignore changes in the robot as a means of measuring environment change and thereby incentivizing exploration (Mendonca et al., 2023).

In machine learning (ML), the distinction between the agent and the environment has been studied under the umbrella of disentangled representations, a long-standing problem in ML (Bengio, 2013; Wang et al., 2024). While most works take an information theoretic approach to disentangled representation learning (Higgins et al., 2017; Kim & Mnih, 2018), some try to take advantage of known structural biases in the data, which is particularly relevant for sequential data, as it usually contains both time-variant and invariant features (Wiskott & Sejnowski, 2002). In video, this allows to disentangle content from motion (Villegas et al., 2017). Denton & Birodkar (2017) explore the insight that some factors are mostly constant throughout a video, while others remain consistent between videos but can change over time, to disentangle content and pose.

Finally, agent-environment disentanglement has seen growing interest in RL, being achieved through attention mechanisms (Choi et al., 2018) or, more commonly, from explicit supervision (Gmelin et al., 2023; Qian et al., 2025; Kim et al., 2025). In Gmelin et al. (2023), the authors demonstrate that learning this distinction allows RL algorithms to achieve better sample-efficiency and performance, serving as our most direct baseline. Vision transformers have also been adopted in the field, as they learn a feature space in which a latent dynamics model can then be trained (Seo et al., 2023). Similarly to our work, some other approaches have combined ideas from both model-free and model-based RL, by augmenting model-free algorithms with prediction-based auxiliary losses (Racanière et al., 2017; Schwarzer et al., 2021). However, these works require learning a full dynamics model and do not consider any distinction between agent and environment.

## 6 CONCLUSIONS

**Analysis and Limitations**   In this work, we studied how motion can be used to disentangle agent information in a self-supervised manner. We integrate our approach with two RL algorithms, demonstrating improved sample-efficiency and performance, particularly in tasks requiring the use of tools. Furthermore, by removing the need for supervision while improving sample-efficiency, our method can be seen as a step towards RL in real-world settings. We keep the generality of our approach, so that it can be used by other researchers on top of other RL algorithms. Some limitations of our work include the need for pixel-wise reconstruction of the scene, an auxiliary task that still generates debate in the field as to whether it imposes the need to model task-irrelevant or impossible to predict information (Hansen et al., 2022; LeCun, 2022). This could be addressed in the future with the use of a contrastive loss, which is computed in feature space and removes the need for reconstruction. The fixed babbling stage is another limitation, as in some tasks the reward immediately shoots up after babbling. In the future, this stage should be made adaptive. Furthermore, our approach still suffers from the characteristic instability of baseline RL algorithms such as DDPG (Islam et al., 2017; Henderson et al., 2018), an issue that hampers development and limits progress in the field. Finally, there is a wall-clock overhead associated with EF (see Appendix A.4). Nevertheless, this penalty is compensated by the gains in sample-efficiency, which in applications like robotics is of greater importance than computational speed.

**Future Work**   Future work directions include further exploration of the adaptation to new body-schemas, in particular how fast the model can adapt to changes and whether it can generalize to previously unseen tools. We also intend to take advantage of the disentangled information to improve modeling of moving objects in the scene. Another potential research direction is to scale experiments to higher resolution images of real-world environments, exploring ways to integrate Transformer architectures into feature learning for RL. Finally, our approach could also be applied to domains outside robotics, such as autonomous driving, where the main difference is that the agent's actions control the optical flow of the observed world and not the agent's body.

ACKNOWLEDGMENTS

This study is in part funded by Lisbon ELLIS Unit, the Center for Responsible AI (PRR), LARSyS FCT funding (DOI: 10.54499/LA/P/0083/2020, 10.54499/UIDP/50009/2020, and 10.54499/UIDB/50009/2020). YD acknowledges the support of UKRI grant EP/Y028732/1, and his RAEng CiET. This work was also supported by grants from NVIDIA and utilized NVIDIA RTX A6000 Ada graphics cards.

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

## A APPENDIX

### A.1 ARCHITECTURE

When extending an algorithm with EF, we preserve the architecture of the visual encoder, as this is shared between the original algorithm and EF. Because our approach requires the introduction of a bottleneck, in DrQv2-EF we introduce an additional average pool and linear projection at the output of the encoder, to reduce the dimensionality of the DrQ-v2 feature vector from 39200 to 2048. For TD-MPC2, the encoder architecture is completely preserved. The recurrent unit is based on the forward dynamics model of TD-MPC2, which consists of an MLP with LayerNorm and Mish activations and has demonstrated strong results in latent space prediction. The decoder is based on DCGAN (Radford et al., 2016).

### A.2 REAL-WORLD ENVIRONMENT

To test the real-world transferability of Ego-Foresight, we perform a qualitative assessment on the predictions on the BAIR dataset (Ebert et al., 2017), which consists of a Sawyer robot randomly pushing a wide range of objects on a table. In Fig. 9, we show two sample predictions from Ego-Foresight. The model succeeds in separating visual information that is part of the robot from the scene, predicting the trajectory of the robot's arm according to its true motion, which shows that the model correctly learned the visuomotor map between actions and vision. The background, including moving objects, is reconstructed in their original position, as encoded in $h_s^{t_c}$. Even so, Ego-Foresight still predicts that some change should happen when the arm passes by an object, and it often blurs objects that predictably would have moved. However, it should be noted that the agent actions and object movement are inherently correlated and therefore cannot be completely disentangled.

We also evaluate the ability to generalize to previously unseen trajectories. To achieve this, we handcraft a previously unseen movement, as shown in Fig. 10. While we display a single handcrafted example, this simple experiment shows that, provided a policy function, the dynamics learned by Ego-Foresight should manage to predict the outcome of sampled trajectories.

In experiments with the BAIR dataset, both the encoder $E$ and the decoder $D$ are based on DCGAN. This enables the addition of U-Net like skip-connections (Ronneberger et al., 2015), which facilitate the reconstruction of fine-grained details and allow the prediction to focus on the agent. These improve the prediction quality without impacting performance and can optionally be used in the simulated experiments too.

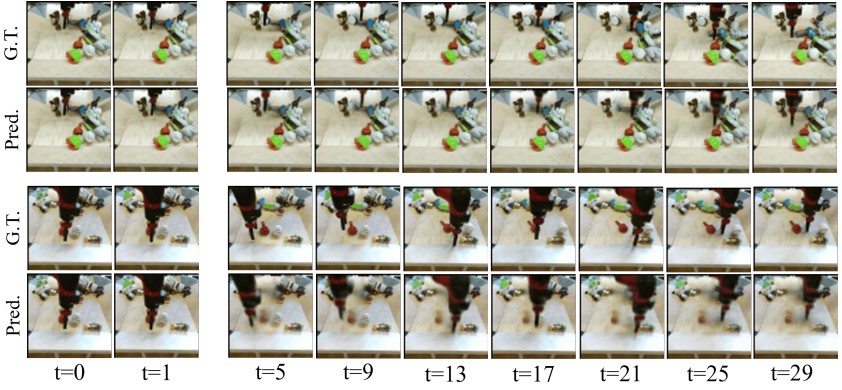

Figure 9: Predictions on the BAIR Dataset.

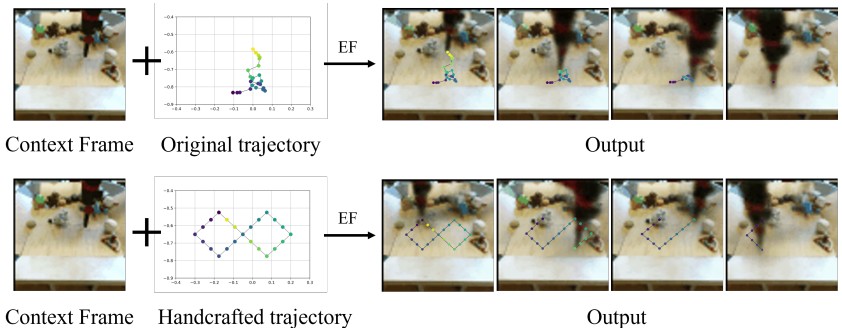

Figure 10: Generation of an unseen trajectory.

## A.3 Supplementary Per-task Results

We present per-task results on the remaining Meta-World tasks that do not require tools in Fig. 11 and the results of the model-free algorithms on DMC described in Section 4.3 in Fig. 12.

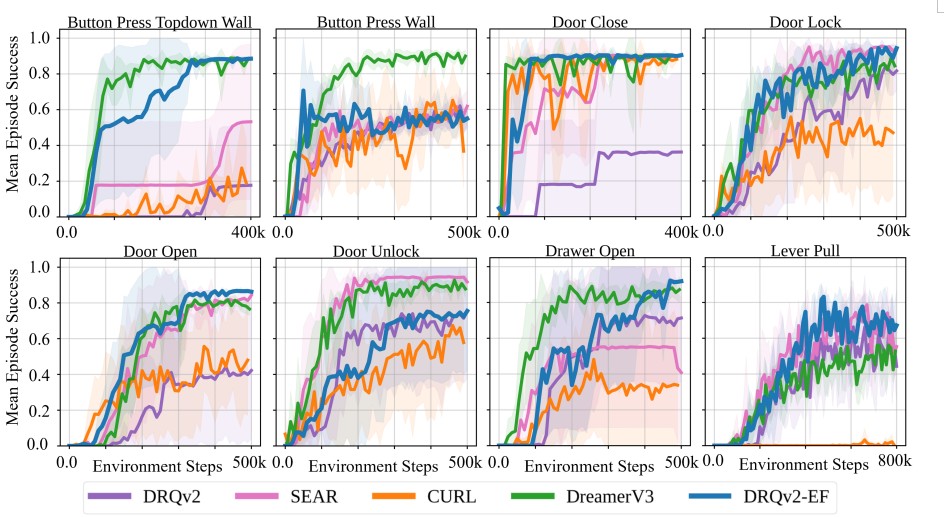

Figure 11: Per-task results on Meta-World tasks not requiring tools.

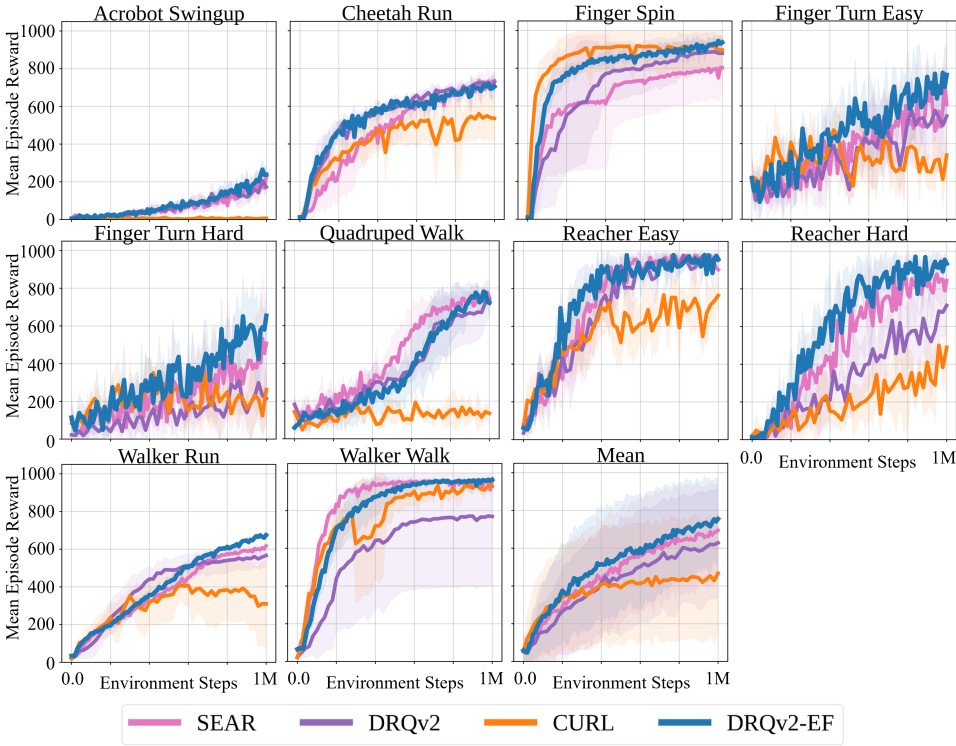

Figure 12: Per-task results on DMC for the model-free RL algorithms.

### A.4  TRAINING DETAILS

We opt to use the official results provided in tables by the authors of each baseline, whenever available, in an effort to save resources. This was the case in the DMC benchmark for DrQ-v2, Dreamer-v3 and TD-MPC2. In all other instances, results are obtained by executing the source code provided by the authors of each work. This is done using NVIDIA RTX A6000 and RTX 6000 Ada GPUs but the models can be run on cheaper hardware too, such as an NVIDIA RTX 4060. The modifications necessary for extending an algorithm with Ego-Foresight are also implemented on top of the original code, without changing the default hyperparameters. A list of the values used for the hyperparameters introduced by Ego-Foresight is provided in Table 1. For the hyperparameters that were part of the original DrQ-v2 and TD-MPC2, we use the default values provided in the respective papers and source code. Similarly, for SEAR and Dreamer-v3, all hyperparameters are the default ones.

In Table 2 we present an analysis of computational cost of each algorithm in terms of number of environment steps taken per second.

Table 1: Default hyperparameters used in DrQv2-EF and TD-MPC2-EF.

| Ego-Foresight Parameters | | |
|---|---|---|
| Parameter | DrQv2-EF | TD-MPC2-EF |
| Hidden dim. | 2048 | 512 |
| $h_s$ dim. | 2016 | 480 |
| $h_a$ dim | 32 | 32 |
| Motor-babbling steps | 25000 | 1250 |
| Prediction horizon (time steps) | 10 | 10 |

Table 2: Environment steps per second obtained on one NVIDIA RTX A6000 using batch size 256 and action repeat 2 on the DeepMind Control benchmark. For Dreamer-v3, the default batch size of 16 was used.

| Algorithm | Env. steps per second |
| --- | --- |
| DrQ-v2 | 155 |
| **DrQv2-EF** | 110 |
| SEAR | 88 |
| CURL | 44 |
| Dreamer-v3 | 41 |
| TD-MPC2 | 28 |
| **TD-MPC2-EF** | 23 |

## A.5   ADDITIONAL VISUALIZATIONS

In Fig. 13 we provide additional visualizations of the learned features on the Hammer task of Meta-World. Feature visualizations are obtained with the method presented in Section 4.2. Each row of Fig. 13 presents a snapshot of the learned features at different training moments, in particular one at the beginning of training, another after the babbling stage and another after the agent has learned to pick up the hammer. From columns (3) and (5) it is possible to visualize how our approach learns an approximation of the true agent mask. Additionally, we note how the hammer begins to be represented together with the robot once it is gripped. Finally, in column (4) we present a thresholded version of (4), allowing a more direct comparison with (5). The pixelated appearance of these frames is a product of the patching step involved in obtaining the visualizations (see Section 4.2). We compute precision and recall between columns (4) and (5) obtaining the pairs: at 5k steps $(0.11, 0.35)$, at 30k steps $(0.32, 0.30)$, at 500k steps $(0.18, 0.17)$. We do note that these values are affected by the size of the patches and by the thresholding value, and are merely indicative.

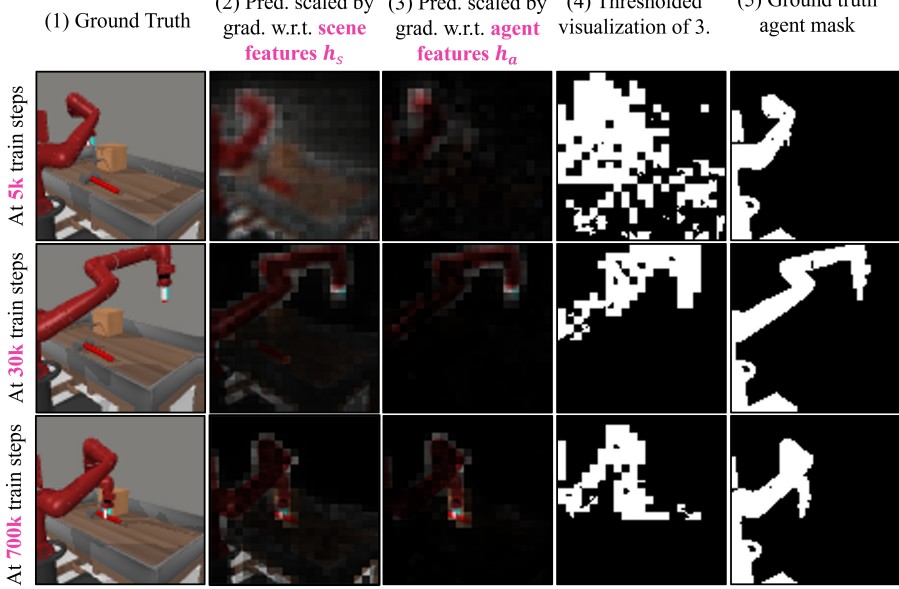

Figure 13: Visualization of the learned features on the Hammer task of Meta-World.

