# OpenReview forum: "Ego-Foresight: Self-supervised Learning of Agent-Aware Representations for Improved RL"
_ICLR.cc/2026/Conference — ICLR 2026 Poster_

### Official Review · Reviewer_hc2R · 2025-10-29

**Soundness:** 3
**Presentation:** 3
**Contribution:** 3
**Rating:** 6
**Confidence:** 3

**Summary:**

This paper proposes Ego-Foresight (EF), a training method that employs an auxiliary task of predicting agent-specific features. The authors show that EF improves sample efficiency when applied to both model-free and model-based RL algorithms. They show that EF lifts limitations of supervised methods for representation disentanglement and enables agents to adapt to changes in their body-schema, such as during tool use.

The authors provide strong motivation for their method and the results are convincing. The contribution of a self-supervised approach for motor prediction is of interest to the RL community. I have some concerns regarding the clarity of the mechanisms underlying EF's performance improvements, but overall, I think this paper should be accepted.

**Strengths:**

EF demonstrates improved sample efficiency when integrated with both model-free and model-based RL algorithms. The performance differences and improved sample efficiency between DRQv2-EF/TD-MPC-EF and their baseline counterparts are convincing across the tasks presented. Additionally, reporting Rliable metrics in addition to the performance curves add robustness to the results since they report statistic uncertainty more rigorously.

EF successfully reconstructs future agent configurations and adapts to changes in body-schema such as tool use (Figure 4). This approach offers advantages over supervised methods, which are constrained by fixed body-schema and cannot adapt to changes such as tool use.

**Weaknesses:**

The authors propose that EF's improvements stem from (i) disentanglement of agent information allowing the RL algorithm to focus on agent control in early training stages and environmental interactions later, and (ii) regularization through imposing predictability in robot movements. However, there seems to be not enough evidence supporting this mechanism, particularly the claim about distinct training phases. While Figure 1 (left) shows some training progression, the predictive ability for agent-specific features appears to improve later in training rather than in the initial stages, which seems inconsistent with the proposed explanation.

I also have a slight concern about the use of Efficiency Normalized Score (ENS). ENS identifies the step at which 95% of maximum performance is reached and measures algorithm performance at that point. This metric may inadvertently favor algorithms with higher performance variance which could reach the threshold faster than more stable learners. So ENS may not be reflecting true sample efficiency accurately.

**Questions:**

**Questions for clarification:**

1. The claim that "predicting the sensory consequences of one's own movement is remarkably simpler, yet equally important" cites Wolpert and Flanagan (2001), but this source doesn't appear to directly support this claim. Could you clarify what you mean by "simpler" and "important" in this context, and provide additional citations if available?
2. You mention "both h_a and h_s have a similar influence in the predicted frame, with solid and static regions such as the background being quickly overfitted by the decoder," and mention the "regularizing effect on learning". Could you elaborate on what constitutes overfitting in this context and how you observe the regularizing effect?
3. You mention "the need for pixel-wise reconstruction of the scene, which could be addressed in the future by exploiting a contrastive loss." Could you elaborate on this idea and how a contrastive loss might address the current limitations?
4. You mention "the characteristic instability of baseline RL algorithms such as DDPG." Did you observe this instability in your experiments?

---

**Minor comments:**

- Figure 4 (left) could be clearer. It's difficult to distinguish the trajectory of the end effector because the end-effector is blurred. The trajectory was more distinguishable in the gif provided in the codebase, so maybe a different angle would be helpful for the static figure.
- The claim that "in terms of ENS, a significant improvement is achieved with the addition of our approach" would be strengthened by adding confidence intervals to Figure 1 to support statistical significance.
- In the ablation study you mention, "In terms of the dimensionality of the agent features h_a, the ablation study shows that larger sizes have a detrimental effect." This wasn't immediately apparent to me because the larger dimensionalities appeared to have similar performance curves to the default value of 32.

---

**Suggestions for improvement (not factored into score):**

1. Could evaluate EF with DreamerV3 to demonstrate generalization across model-based algorithms.
2. Include additional analysis demonstrating the proposed distinct phases where the model learns agent-specific versus environment-specific features.
3. Compare the performance boost from tool use versus no tool use, and compare this difference with the difference given by the supervised baseline (SEAR). Given your claims about supervised approaches being non-adaptable to body-schema changes, EF should demonstrate better performance gains on tool-use tasks compared to SEAR.
4. The ablation study performance curves on the Door Open task show high variance with similar results across conditions. Could consider conducting ablations on tasks with larger differences, such as bin picking or peg insertion, to better distinguish the impact of different hyperparameters.
5. You mention that sample efficiency is more important than computational efficiency, but it would still be helpful to quantify the wall-clock time tradeoff of EF.

---

> ### Author Response · Authors · 2025-11-19
> **Response to Reviewer hc2R**
>
> We thank the reviewer for the insightful comments, detailed analysis of our work and suggestions for improvement. In the following we try to clarify your questions and continue discussion on the raised weaknesses.
>
> **Weaknesses:**
>
> **W1 - Training Phases.** Thank you for your comment, we have tried to provide more clarification on this point in the paper, at the end of section 3.3. In our experience, the predictive ability for agent features improves mostly during the babbling stage at the beginning, which compels the agent to perform a wide range of motions, exploring possible agent configurations and filling the Replay Buffer with a diverse set of data. After this stage, actions begin to be sampled from the policy, leading to more frequent task-oriented actions, at which point observations become less diverse and is more difficult to learn the mapping between action-sequence and agent configuration. Still, having gone through the babbling stage in the beginning, when the task-oriented stage begins, the encoder is already quite good at disentangling agent features, with the intuition being that it should help maximize rewards related with agent control (e.g. get closer to target object) faster.
>
> In Figure 1 in particular, we produce the second visualization at 30k steps, which after 25k babbling steps and in a budget of 500k steps could still be considered the initial stages of training.
>
> In tasks requiring the use of tools, there is indeed a need for adaptation later in training, as the agent typically takes some time to learn to pick-up the tools. We refer to our reply to Q5 of reviewer eUBc for more discussion on this topic.
>
> **W2 - Efficiency Normalized Score.** We agree that every metric has shortcomings or fails to convey the full picture. Because measuring ENS at a single threshold could make it more susceptible to favoring algorithms with higher variance, we used 3 different thresholds (85, 90 and 95%) and averaged the results to obtain a more robust ENS  score. We hope this can increase the fairness of the ENS in comparing the results. For better transparency and clarity we decided to provide the per-task results.
>
> **Questions:**
>
> **Q1 - Motor Prediction/Importante.** With the mentioned reference we intended to direct readers to the concept of motor prediction. We now notice that this reference does not discuss the importance of different types of stimulus, something that is discussed by some of the same authors in other papers, for example in [1]:
>
> On self-produced stimuli being less important the authors open the Introduction with:
>
> *“Our sensory systems are constantly bombarded by a multitude of sensory stimuli, from which we must extract the few stimuli that correspond to important changes within the environment. One class of stimuli that are in most circumstances unimportant and can be discarded are those that arise as a necessary consequence of our own motor actions.”*
>
> On the other hand, on external stimuli being of higher importance they say for example:
>
> *“We propose that prediction-based modulation acts as a filter on incoming sensory signals that can enhance the afference-to-reafference ratio (…).* This modulation of incoming sensory
> input might have the effect of accentuating features of importance (for example, those due to external events)”
>
> These phenomena are discussed by other authors such as Andy Clark in [2], who provides more examples in animals such as fish. (Chapter 4, section 4.2)
>
> We do note that these references served as inspiration to guide our work and that we do not claim to implement the mechanisms discussed in them.
>
> We notice that we mention motor-prediction as equally important, which we meant in the sense that it allows external signals to become more salient. We have rephrased this sentence to make this point more clear and added the additional reference from Blakemore, Frith & Wolpert.
>
> *[1] Blakemore, S. J., Frith, C. D., & Wolpert, D. M. (1999). Spatio-temporal prediction modulates the perception of self-produced stimuli. Journal of cognitive neuroscience, 11(5), 551-559.*
>
> *[2] Clark, A. (2015). Surfing uncertainty: Prediction, action, and the embodied mind. Oxford University Press.*
>
> *(continues in next comment due to character limit)*

---

> > ### Author Response · Authors · 2025-11-19
> > **Response to Reviewer hc2R (continuation)**
> >
> > **Q2 - Overfitting and Regularizing effect.** By overfitting we mean that because the background is the same across all observations (static), the decoder can learn to always produce the same pixel values in the background regions, and the feature representation is therefore relieved from having to encode background information. Hence, the gradient of background image patches w.r.t. the features is close to zero (image 1 left). Intuitively, the decoder can “memorize” the pixel values of the background and static regions of the environment.
> >
> > We refer to regularization as adding a loss term that is independent of reward and does not directly optimize towards solving the RL task. In the original actor-critic, the encoder is updated in the critic update step, meaning that it is trained to extract features that are predictive of value. We find that adding the additional term inducing the extracted features to also be predictable of agent motion regularizes the learned features. We observe this in the ablation of $\beta$, where performance peaks at $\beta=1$ and is lower for smaller and higher values of $\beta$.
> >
> > We’ve added clarification on this issue in Section 3.3. of the document.
> >
> > **Q3 - Contrastive Loss.** The use of reconstruction vs. learning in latent space is a topic of active discussion. Some prior work has claimed that reconstruction may impose the need to model everything in the environment, including task-irrelevant information [1] and fine-grained information that could be impossible to predict [2]. On the other hand, some authors have stuck with the use of decoders and pixel level reconstruction [3], with for example the recently released Dreamer-v4 [4] leveraging Transformer architectures for reconstruction.
> >
> > For this reason, while we haven’t worked out the details of how a contrastive loss could be used to learn agent-aware features, we see it as a possible avenue of research to improve our method, by removing the need to model irrelevant parts of the environment. A possible inspiration for this work would be CURL [1].
> >
> > We have added more detail on this topic in the Analysis and Limitations section.
> >
> > *[1] Hansen, N., Wang, X., & Su, H. (2022). Temporal difference learning for model predictive control. International Conference on Machine Learning.*
> >
> > *[2] LeCun, Y. (2022). A path towards autonomous machine intelligence version 0.9. 2, 2022-06-27. *Open Review*, *62*(1), 1-62.*
> >
> > *[3] Xiang, J., Gu, Y., Liu, Z., Feng, Z., Gao, Q., Hu, Y., et al.. (2025). PAN: A World Model for General, Interactable, and Long-Horizon World Simulation. *arXiv preprint arXiv:2511.09057*.*
> >
> > *[4] Hafner, D., Yan, W., & Lillicrap, T. (2025). Training agents inside of scalable world models. *arXiv preprint arXiv:2509.24527*.*
> >
> > *[5] Laskin, M., Srinivas, A., & Abbeel, P. (2020, November). Curl: Contrastive unsupervised representations for reinforcement learning. In *International conference on machine learning* (pp. 5639-5650). PMLR.*
> >
> > **Q4 - Instability.** We do observe variance across seeds, which we consider one of the most complicating factors of developing new RL algorithms, a topic that has been explored in multiple works [1] [2] [3]. This means that the effects of a given modification to an RL architecture/approach can only be interpreted after repeating the experiments for multiple random seeds, which significantly slows down progress.
> >
> > We try to combat this issue by providing results averaged across seeds, with aggregate metrics and RLiable scores, which were proposed in [3] as a more robust method for reporting RL results.
> >
> > *[1] Riashat Islam, PeterHenderson, MaziarGomrokchi, andDoina Precup. Reproducibility of benchmarked deep reinforcement learning tasks for continuous control. arXiv preprint arXiv:1708.04133,2017*
> >
> > *[2] Henderson, P., Islam, R., Bachman, P., Pineau, J., Precup, D., & Meger, D. (2018, April). Deep reinforcement learning that matters. In *Proceedings of the AAAI conference on artificial intelligence* (Vol. 32, No. 1).*
> >
> > *[3] Rishabh Agarwal, Max Schwarzer, Pablo Samuel Castro, Aaron C Courville, and Marc Bellemare. Deep reinforcement learning at the edge of the statistical precipice. Proc. of the Conf. on Neural Information Processing Systems (NIPS), 34:29304–29320, 2021*
> >
> > *(continues in next comment*)

---

> > > ### Author Response · Authors · 2025-11-19
> > > **Response to Reviewer hc2R (continuation)**
> > >
> > > **Minor comments:**
> > >
> > > 1. **Figure 4:** we have increased the size of this Figure and added new sequence with more steps, in order to allow the motion of the robot to be more perceptible. The blurriness of the image is a consequence of using a model trained on an horizon of 10 steps to predict for longer horizons, as well as the option for a light convolutional encoder-decoder. We have also added to the discussion about this figure in the text. We hope the changes have made the figure more clear.
> > > 2. **Confidence intervals**: we have added a version of Figure 1 with standard deviation bars in Figure 14 (Appendix) and will replace it in Figure 1 if you find that it improves the quality of the paper. These were not included in the original submission because standard deviation would be computed across tasks with different levels of difficulty and therefore performance at congergence. This results in very high standard deviation which we found could be misleading. For this reason, we followed the approach of the literature, in particular Dreamer-v3 and TD-MPC2 in not reporting standard deviation in the aggregate bar plots.
> > > 3. **Dimensiontality of $\boldsymbol{h}_a$**: for sizes 32 and 64 performance converges to around 4300 vs. 4000 and below for values below and above. We refer to the discussion with reviewer Y4MG about the relevance of changes in performance in this magnitude.
> > >
> > > **Suggestions for improvement:**
> > >
> > > 1. **Evaluating EF with Dreamer-v3**: we again refer to the discussion with reviewer Y4MG. While conducting our work, we identified TD-MPC2 and Dreamer-v3 as the leading model-based approaches in the literature. We chose to modify only one of them due to time constraints and not believing that integrating EF with multiple model-based methos would change the conclusions of our paper. Nevertheless, we agree that these would be interesting results, which we hope to obtain in the future.
> > > 2. **Additional analysis on distinct phases**: We have added an appendix with additional visualizations showing learned features at different moments during training.
> > > 3. **Tool-use vs. no-tool**: In light of this comment we have added an additional plot to Figure 1 with results for the subset of tasks that do not require tools.
> > > 4. **Ablation on Door Open**: for reasoning on the choice of this task for the ablations we refer to our reply to reviewer eUBc.
> > > 5. **Wall-clock**: we have added an analysis of the computational requirements in Appendix A.4.

---

> > > > ### Comment · Reviewer_hc2R · 2025-11-23
> > > >
> > > > I appreciate the detailed response from the authors. The elaboration on training stages does help clarify what is learned at each phase of training. I also appreciate the discussion and references related to motor prediction and contrastive loss. The updated version of Figure 1 is better since it clearly shows that the improvement of EF on tool use is higher that improvement on non-tool use tasks.
> > > >
> > > > Reviewer Y4MG points out the difference in convergence values for horizon length, but I don't think the different is too significant since the standard error ranges are so large. That is, the authors are correct in concluding that horizon length doesn't impact the performance of EF too greatly. My original point was that the differences in the ablation studies weren't too notable since the standard error ranges are large. However, this is a minor point that doesn't affect my overall assessment of the work.
> > > >
> > > > With regard to ENS, I am not convinced that using multiple thresholds improves the suitability of this metric since high variance algorithms would have an advantage at all the thresholds. However, the training curves and Rliable metrics are sufficient to support the author's claims that EF generally improves sample efficiency over its base algorithm. Therefore, I continue leaning toward acceptance.

---

### Official Review · Reviewer_Y4MG · 2025-10-30

**Soundness:** 2
**Presentation:** 2
**Contribution:** 2
**Rating:** 4
**Confidence:** 4

**Summary:**

This paper proposes Ego-Foresight (EF), a self-supervised method, where the core idea is to disentangle the agent’s representation from the environment’s representation by leveraging motor prediction. The model learns to separate a feature vector h into agent-specific features ha  and scene-specific features hs. A recurrent model predicts future agent features based on unrolled action sequences, while the scene features hs are fast-forwarded from the context frame. The predicted agent features and the original scene features are then used to reconstruct the predicted frame to train the encoder. This method avoids the need for explicit supervision required by prior work like SEAR. The authors demonstrate that adding EF as an auxiliary loss to model-free (DrQ-v2) and model-based (TD-MPC2) algorithms improves performance and sample efficiency on Meta-World and DMC tasks.

**Strengths:**

- The core idea of using motor-prediction as a self-supervised signal for agent-environment disentanglement is interesting. It provides an alternative to supervised methods that rely on segmentation masks, which are often unavailable in real-world settings.
- The proposed EF module is model-agnostic and can be integrated as an auxiliary loss into both model-free and model-based RL algorithms, as demonstrated by its successful application to DrQ-v2 and TD-MPC2.
- The method shows clear improvements in sample efficiency and asymptotic performance against strong baselines (DrQ-v2, TD-MPC2) and is competitive with its supervised counterpart (SEAR) on several tasks.

**Weaknesses:**

- Figure 4 (“Door Open” and "Hammer" task) seems to be central to the claims made in the paper and tries to show that the door is part of the environment whereas the hammer becomes part of the body schema. However, the distinction is not so clear to me. the hammer has to be picked up first, just like the agent has to grip the door handle. Once the contact is made the door could be moved to and fro just like the hammer. The illustrations in the figure also does not really help. It seems like the gripper disappears in the door opening task, i can't even see whether the door is open or close in the picture...

- The authors state that “horizon length does not significantly impact the results”. However, the “Prediction Horizon” graph in Figure 8 clearly shows that H=10 and H=40 (peak reward ~4500) perform significantly better than H=5 and H=20 (peak reward ~4000). A ~500 reward difference (or ~12.5%) is not insignificant. It is similiar to the difference between β=0  (no EF module) and other β where it is claimed that the difference is important. This difference in treatment of the same kind of error difference seems concerning wrt to the paper's claims.

 Minor issues:

- Presentation: The paper's structure could be tightened. Section 2 ("Background") is a single, long subsection that could be broken up. Similarly, for Section 4.1 ("Experimental Setup"), the special subtitle "Environments" is not necessary.

- Typo: On Line 238, the feature vector is referred to as $h_{t_c}$. To be consistent with the paper's notation defined on Line 191, this should be $h^{t_c}$.

- Clarity: In Figure 2, the "action" label is a bit vague. It seems strange to use a single variable $x$ to represent the observation, and use a vector $[a^0, \cdots, a^M]$ to represent the action.

**Questions:**

## Questions
1. Following Weakness #1: Could the authors please clarify the role of $h_s$? Is it intended to be static? Why does the gripper disappear in the Figure 4 prediction, and what does this imply about what $h_s$ is actually encoding?

2. Following Weakness #2: Could the authors justify their claim that a ~500 reward gap in the horizon ablation is "not significant"? It appears to be one of the more significant factors in the study.

3. The qualitative results in Figure 4 show predictions for relatively small movements. How does the model's predictive quality and disentanglement hold up when the agent performs large-scale actions or when predicting over a longer horizon?

4. in Fig. 5 and 6, the authors do not integrate their approach into the Dreamer architecture, but only in the other two approaches. Why is that?

---

> ### Author Response · Authors · 2025-11-19
> **Response to Reviewer Y4MG**
>
> We thank the reviewer for the feedback. In light of your comments, we have modified the paper with new clarifications and better presentation. Below we follow up on the weaknesses and questions.
>
> **Weaknesses:**
>
> **W1 - Door Open vs. Hammer.** This is a really interesting point, as the line between what can be integrated into the the robot’s embodiment can indeed be blurry. We believe this is mostly dictated by how much the actions of the robot determine the motion of the tool/object. When the task requires gripping an object, this will be integrated into the feature representation.
>
> In the hammer vs. door case: the motion of the hammer is completely determined by the motion of the robot once the hammer is gripped; the same is not true for the door since after the robot makes contact with it,  the movements of the door and of the robot diverge, meaning that the door cannot be predicted from the sequence of actions of the robot.
>
> For example, in some episodes the robot might bat the door with force X (opening it and completing the task) and then move in another direction as the door continues to open. In another episode the robot might bat the door with a force Y, (still completing the task) and move in a new direction after the contact. The lack of consistency between the actions of the robot and the motion of the door make it unfeasible for the motion of the door to be predicted. It is then reconstructed in the configuration observed in the context steps, encoded in $\boldsymbol{h}_s$.
>
> This intuition is inspired by [1], where the authors study neuron activations during tool-use and the encoding of body-schema.
>
> We also note that in the Figure 4, the model was trained to predict 10 steps into the future, while the evaluation is produced for a longer horizon which together with the use of a simple encoder-decoder leads to some blurriness in the predictions.
>
> We have added a new sequence with more steps to make movement more discernable and increased the size of the figure. We have also added to the analysis of Figure 4 in Section 4.2.
>
> *[1] Angelo Maravita and Atsushi Iriki. Tools for the body (schema). Trends in cognitive sciences, 8 (2): 79–86, 2004*
>
> **W2, W4, Q2 - Horizon ablation.** We thank the reviewer for this remark, as the text is indeed misleading regarding what we intended to convey. We have updated the text accordingly in the Ablation Study section.
>
> We do agree that a difference of 500 in reward is a significant gap. We intended to convey that we did not observe a **consistent**  impact of the horizon length in the results, and this is what should have been transmitted in the writing.
>
> When compared to the (unmodified) DrQ-v2 (in Figure 11), all values tested for the horizon led to an improvement, hence our assertion that results are good even for shorter horizons. Still, as pointed out by the reviewer, there is a significant difference between the values 10 & 40 vs. 5 & 20. When choosing between the best values we opt for 10, as it incurs a lower computational penalty.
>
> **W3 - Presentation.** We thank you for your suggestions. We have modified the paper accordingly.
>
> **W5 - Correction in Figure 2.** We have modified Figure 2 to represent both actions and observations with a single variable in bold as in the definition of $e_i$.
>
> **Questions:**
>
> **Q1 - Clarification on $\boldsymbol{h}_s$.** The role of the $\boldsymbol{h}_s$ is to encode the visual content of the whole scene (both agent and remaining parts of the environment), and has a role both in solving the RL task and in the auxiliary prediction objective.
>
> For reconstruction of the future frame at step $t_k$, we fast forward the $\boldsymbol{h}_s^{t_c}$ vector obtained with the context input, which holds information about the environment and the configuration of the agent at $t_c$. This allows the decoder to receive information about the content of the scene, which shouldn’t be available in $\boldsymbol{h}_a$ but is necessary for reconstruction. It also relieves the recurrent block from having to encode scene information (as it is already provided to the decoder by $\boldsymbol{h}_s$), allowing it to encode agent information instead. The feature components corresponding to $\boldsymbol{h}_a$ are then responsible for providing information to the decoder about the configuration of the agent at step $t_k$, allowing reconstruction of the future frame. Because scene information comes from $t_c$, the reconstructed frames tend to be close to $\boldsymbol{x}^{t_c}$ except for the configuration of the agent.
>
> This effect is particularly visible in the BAIR experiments of Figure 9, where more objects are present in the scene. For example, in the G.T. sequence at the top, the blue gun is displaced but in the predicted frame it is reconstructed in its original position, with only the agent in the future position.
>
> *(continues in next comment due to character limits)*

---

> > ### Author Response · Authors · 2025-11-19
> > **Response to Reviewer Y4MG (continuation)**
> >
> > **Q1 (cont.)** Regarding blurriness in the gripper and other parts of the predictions, we attribute this to the fact that the model is trained with an horizon of 10 steps, and in the figure is used to predict for longer horizons.
> >
> > We have modified the text in Section 3.2 with the intention of making the role of $\boldsymbol{h}_s$ more clear.
> >
> > **Q3 - Predictive Quality.** In our experience, this depends on the prediction horizon used during training and on the data collected by the agent. We find the model robust to large scale movements which are more common during the babbling stage. Similarly, the model is capable of predicting a few steps after the training horizon, but starts to diverge when these go too far into the future. This can be seen in Figure 4, where we show predictions until time step 28 for a model trained on an horizon of 10 frames.
> >
> > We find that an horizon of 10 steps is enough for EF to learn to disentangle agent information, which in the end is the goal of the auxiliary predictive objective and not necessarily the predictive quality. This interpretation is supported quantitatively by the horizon ablation test and qualitatively by the visualization of figure 1, which is obtained with an horizon of 10.
> >
> > Nevertheless, we believe H=10 could be considered a long horizon when compared with algorithms such as TD-MPC2, which uses a default horizon of 3 steps.
> >
> > **Q4 - Integration with Dreamer-v3.** While conducting this work, we prioritized the integration of our approach to DrQ-v2, in which we developed the bulk of the experiments and then tried to demonstrate the ability to generalize the approach to other algorithms, in particular model-based ones. We identified TD-MPC2 and Dreamer-v3 as the main model-based approaches in the literature and opted to modify only one of them both due to time constraints and for believing that integrating our approach on both algorithms would not significantly change the conclusions of our work.
> >
> > In the end, the choice of which model-based algorithm to modify came down to practical aspects such as TD-MPC2 being implemented in torch, a framework with which the authors were already familiar, while the official Dreamer-v3 implementation uses JAX.

---

### Official Review · Reviewer_Nv8q · 2025-10-31

**Soundness:** 4
**Presentation:** 3
**Contribution:** 3
**Rating:** 8
**Confidence:** 3

**Summary:**

This work proposes idea of self-supervised learning of consequences of RL agent's actions in an environment for inducing agent-aware representations. The motivation behind the work is grounded in how humans foresee the effects of their actions while executing them which allows them to perform the tasks more effectively. The paper proposes use of convolutional encoder-decoder architecture that predicts future frames given the visual features and sequence of planned actions. A self-supervised loss is applied on the predictions that allows us to learn visual features that enable easy prediction of the future, and hence instill agent-awareness. The authors extend pre-existing RL algorithms in robotics with proposed ego-foresight loss that allows for improved performance.

**Strengths:**

1. The paper is very well written with the introduction making connection of the technique with human locomotion and various other foresight instances.
2. The approach proposed in this work seems intuitive and should be easy to integrate into existing RL pipelines learning from visual features.
3. The results demonstrate consistent improvement when the technique is used in conjunction with prior frameworks.
4. The choice of the research questions studied is apt for this work and the paper goes into detail answering each of them.

Overall, I find the quality of the write-up, the experiments and the results high and recommend accepting the work.

**Weaknesses:**

1. The paper uses recurrent neural networks, while the overall efficacy of transformers in sequence generation is well-proven in the domains of natural language and vision.
2. Episode partitioning uses a fixed constant H for defining the future window. This choice could have been made more dynamic to remove any bias due to burden of predicting next H frames every time.

**Questions:**

Are these results also applicable in non-robotics environments like standard control tasks (e.g. CartPole) in RL? It would be nice to have an auxiliary loss that allows the neural network to predict the next states to align the representations used for deciding the policy. It is only a minor suggestion.

---

> ### Author Response · Authors · 2025-11-19
> **Response to Reviewer Nv8q**
>
> We thank the reviewers for their valuable comments and follow-up on the discussion below.
>
> **Weaknesses:**
>
> **W1 - Use of Transformers.** We do agree that exploring models with greater representational power could lead to improved results, especially if scalling up to applications involving real-world scenarios with high resolution observations. Still, we note that adoption of Transformers in RL still faces some limitations [1], and we hope to come back to this issue in the future. In this work, the choice for a recurrent neural network was based on demonstrated success of the same architecture in TD-MPC2.
>
> We have added this to the paper as a possible avenue of Future Work.
>
> *[1] Hu, S., Shen, L., Zhang, Y., Chen, Y., & Tao, D. (2024). On transforming reinforcement learning with transformers: The development trajectory. *IEEE Transactions on Pattern Analysis and Machine Intelligence*, *46*(12), 8580-8599.*
>
> **W2 - Bias due to fixed H.** Thank you for pointing out this issue. In our approach, we do define a fixed horizon for prediction, as per common practice in other algorithms using prediction, such as TD-MPC2. In trying to mitigate the bias, at each training update we randomly sample one time step $t_k$ inside the fixed horizon, where prediction stops, the frame is reconstructed and $\mathcal{L}_{ef}$ is computed. Hence the the reconstruction step varies inside the fixed horizon from one iteration to the next.
> This approach has the added benefit of reducing the computational demands of our method.
>
> **Questions:**
>
> **Q1 - Standard Control Tasks.** Yes, the method also works for standard control tasks such as Cartpole. In this particular case, the agent is identified as the pole, which corresponds to visual information that is directly controllable by the actions of the agent.
>
> We hypothesize that our method could be applied to any state representation that can be decomposed into controllable and non-cotrollable components. In the Future Work we refer autonomous driving as a possible application scenario outside of robotics where the actions control the optical flow of the observed environment.

---

### Official Review · Reviewer_eUBc · 2025-11-01

**Soundness:** 2
**Presentation:** 3
**Contribution:** 2
**Rating:** 4
**Confidence:** 4

**Summary:**

This submission proposed Ego-Foresight (EF), a self-supervised method using motion prediction to disentangle agent information from environment, aiming at improving sample efficiency for deep RL. The method uses an encoder-decoder architecture with a recurrent module that predicts future agent configurations from actions, with a bottleneck forcing focus on agent-specific features. The authors integrate EF with DrQ-v2 and TD-MPC2, and compare with SEAR and Dreamer on Meta-World and DMC benchmarks, showing improved sample efficiency particularly on tasks requiring tool use.

**Strengths:**

1. The proposed self-supervised framework for learning agent-aware representation, as opposed to supervised training that requires agent masks in previous work.
2. The proposed method can be integrated with both model-free (DrQ-v2) and model-based (TD-MPC2) algorithms, showing flexibility.
3. The evaluation is comprehensively done on 26 tasks across two benchmarks and baseline methods, including Dreamer (model-based), SEAR (previous supervised agent-aware learning method). Results are presented with proper statistical analysis with RLiable metrics, and the ablation studies are useful for dissecting the proposed model.
4. The paper is generally well-written with good use of figures, including the model diagram.

**Weaknesses:**

1. The results on performance improvement does not seem to be consistent across the tasks. In figures 5 and 6, SEAR and Dreamer seem to perform equally well, and sometimes even better, than EF. This performance gap seems to be more prominent in figures 11 and 12. This inconsistency is not adequately explained. Also, for efficiency normalized score presented in figure 1, could you please provide standard deviation/ errors to properly assess the improvement?
2. Since the proposed EF algorithm is a self-supervised algorithm for image based deep RL, why are other self-supervised image-based RL algorithms not included as comparison baselines, such as CURL (Srinivas, Laskin, et al, 2020) and SPR (Schwarzer, Anand, et al, 2021)? This and the performance inconsistency are perhaps the main weaknesses of the current draft.
3. The ablation study is performed on the DoorOpen task, which is not part of the MetaWorld tasks that require tool use as presented in figure 5. Since you highlighted EF should be better in tasks that require tool use, why is the ablation study done on DoorOpen?
4. As learning efficiency is emphasized, it’d be good to provide comparisons on computational overhead (wall-clock time, memory, etc.), which is mentioned but not quantified.
5. Maybe I missed something, but in terms of the algorithm (neuroscience inspiration aside), Intuitively, why would learning disentangled representation lead to improved sample efficiency?

**Questions:**

1. Can you explain more what Fig 4 is implying? Perhaps using the same visualization for Fig 1 to highlight what’s actually being learned by the agent feature?
2. In the ablation study, larger agent feature dimension seems to have a detrimental effect, which is attributed to “reducing the ability to disentangle agent information.” Might an alternative explanation be just because when the total feature dimension is too large, the policy is harder to learn as the state space is now larger?
3. For tasks where EF doesn't help (e.g., Walker tasks), what is the failure mode? Can you characterize when EF is expected to help?
4. How good is the learned agent feature? Since EF is proposed to disentangle agent from the environment, can you quantify how much overlap there is between the learned agent feature with the supervised agent masks if available in some tasks?
5. Tool embodiment is mentioned as an interesting capability, and is perhaps when EF might be helpful the most. Does this adaptation happen within a session? Say in the hammer task, does the agent feature starts to include the hammer only after the hammer is being picked up? Or does the agent feature already include the hammer from the beginning?

---

> ### Author Response · Authors · 2025-11-19
> **Response to reviewer eUBc**
>
> We thank the reviewer for their valuable feedback. We have addressed your points and improved the paper accordingly. Bellow we address the weaknesses individually and provide some clarification on the questions.
>
> **Weaknesses:**
>
> **W1 - Inconsistent performance in per-task resuls.** Indeed, performance results in these benchmarks tend to vary across tasks, making it difficult to predict performance on a per-task basis. We see this variability as a consequence of the sensibility of current RL algorithms to changes in reward functions (e.g. Door-Open & Door-Close have quite different reward functions [1]), as well the number of degrees of freedom and dynamics of the agent (in DMC). While this leads to variability in performance, it also tests robustness of the algorithms.
> In trying to provide a clear picture of our results, we included both aggregate metrics (ENS and RLiable) and per-task results, for greater transparency, following the approach of other works such as Dreamer-v3 and TD-MPC2. We note that in these works the inconsistency in per-task results can also be observed, for example in Appendix P of the Dreamer-v3 paper (visual control on DMC), where the proposed method is matched on 5 and outperformed on 6 of the 22 tasks, and in FIgure 10 of the TD-MPC2 paper (Visual RL on DMC) where the proposed algorithm is matched in 4 and outperformed in 3 of the 10 tasks.
>
> In an attempt to improve the paper, we've added more discussion on this issue in Section 4.3 (Reinforcement Learning Results paragraph), as well as a plot with the mean over all the DMC tasks in Figures 6 and 12 (because the Meta-World tasks were run for different numbers of steps, the same aggregate plot is not possible in this benchmark)
>
> Regarding standard deviation, we present a version of Figure 1 with standard deviation bars in Figure 14 (Appendix). We’ll move this to Figure 13 if you consider it improves the quality of the paper. In Figure 1, standard deviation was not included in the initial submissionn because this would be taken across tasks. Since there’s a high variability in the difficulty of the tasks, some samples converge to high reward values and others to lower ones, resulting in a high standard deviation on all tested algorithms that we found could be measleading. We again followed the approach of TD-MPC2 and Dreamer-v3 in not presenting standard deviation in the aggregate bar plots.
>
> *[1] Tianhe Yu, Deirdre Quillen, Zhanpeng He,Ryan Julian,Karol Hausman, Chelsea Finn, and Sergey Levine .Meta-world: A benchmark and evaluation for multi-task and meta reinforcement learning. In Conference on Robot Learning (CoRL), pp.1094–1100. PMLR, 2020.*
>
>
> **W2 - Additional Self-Supervised Baselines.** We took steps to add CURL as an additional baseline. For the DMC benchmark, we reuse the results provided in the official Dreamer-v3 repository. For Meta-World results we executed the code of the official repository. In these, we use the hyperparameters indicated in the paper except for batch-size and action-repeat, which for fairness we match with the ones used for the other algorithms. Results show that our approach overperforms this self-supervised alternative.
>
> **W3 - Ablation study on the Dooe Open task.** There were two main reasons for the ablation study to be done on DoorOpen.
> From a practical standpoint, while developing our method we focused mainly on tasks that require less steps to be solved, which for the most part are the ones not involving tool-use. In our approach we do not distinguish between types of task, and the goal was to have a single set of hyperparameters for all the tasks.
>
> Still, the strongest factor in opting for a non-tool task for the ablation tests was that optimizing our hyperparameters for tasks requiring tools could detract from the possible conclusion (at the time an hypothesis) that we'd observe a wider performance gap in those tasks, as the algorithm would be fine-tuned for them. I.e., if the hyperparameters were chosen for tool tasks, then it would be expected that performance would be better for those tasks. As such, before running the final experiments we opted for ablation tests on a non-tool task.
>
> **W4 - Computational overhead.** Following your observation we have added details on the computational overhead in Appendix A.4 (Training Details).
>
> *(continues in next comment due to character limit)*

---

> > ### Author Response · Authors · 2025-11-19
> > **Response to reviewer eUBc (continuation)**
> >
> > **W5 - Intuition.** The intuition for why disentangling agent features should lead to improved sample-efficiency is related with the fact that, in most tasks, the first challenge is to learn control of the agent, for which the agent can usually obtain some reward (e.g. distance between gripper and target object, or forward locomotion). Hence, having a better feature representation for the agent’s embodiment should help in the initial stages of training, especially in the absence of state information. Overall, this should also mean that the agent can identify other sources of reward not related with its own control faster, such as spinning an object or hammering a nail.
> > The same intuition is present in the SEAR paper [2].
> >
> > *[2] Kevin Gmelin, Shikhar Bahl, Russell Mendonca, and Deepak Pathak. Efficient RL via disentanled environment and agent representations. In International Conference on Machine Learning (ICML), volume 202 of Proceedings of Machine Learning Research, pp. 11525–11545. PMLR, 2023*
> >
> > **Questions:**
> >
> > **Q1 - Clarification on Figure 4.** We refer to the reply to W1 of reviewer Y4MG for additional detail.
> > With Figure 4 we intend to provide a qualitative assessment of the ability of our model to:
> >
> > 1. Correctly model the motion of the agent
> > 2. Disentangle agent information, which can be observed in the Door Open task
> > 3. Show that when a tool is gripped, EF learns to encode it into the representation as part of the body of the robot.
> >
> > Following your suggestion, we have added an additional visualization on the hammer task, with the intent of better highlighting what is learned (see Appendix A.5). We have also increased the number of steps in Figure 4 to allow more movement to be shown, increased the size and added more disccusion about the figure to the paper, in order to better explain what is implied.
> >
> > **Q2 - Feature dimension size.** Yes, we believe that too large of a state-space would indeed have a detrimental effect in policy learning. The reason we didn’t consider this possibility in the paper is that this size is actually fixed, as increasing the size of $\boldsymbol{h}_a$ reduces the size of $\boldsymbol{h}_s$ (details in section 3.2). Still, this hints at another explanation that going over the dimension necessary to represent the state-space of the agent then removes representational capacity to $\boldsymbol{h}_s$.
> > We've added further discussion on this issue in the paper.
> >
> > **Q3 - Failure mode.** As mentioned with regards to W1, it is difficult to assess performance on a per-task basis due to sensitivity of the algorithms. For example in Fig. 6 EF doesn't improve vs. the original algorithm in the Walker Run task but does improve in Walker Walk and in Cheetah Run. The best correlation with performance that we identified was in tasks requiring tools.
> >
> > **Q4 - Overlap with supervised masks.** This is an interesting analysis. We have added a new figure in Appendix A.5 where we threshold the visualization of learned features to allow comparison with the ground truth masks. We also provide precision and recall values between the two. One shortcoming of the analysis is that the feature visualizations are obtained by first patching the predicted frame, resulting in a binary image with large white patches, which then affects the precision and recall values. Using smaller patches would improve the precision and recall values.
> >
> > **Q5 - Adaptation to tools.** We see these as some of the most interesting questions that result of our work, and included some in the Future Work. The adaptation happens within a session. In the initial stages, the robot doesn’t know how to pick up the hammer, and therefore the agent’s features only encode the robot itself. Once the robot learns to pick up the hammer, EF learns to integrate it into the agent’s representation after a few more training steps, something that we observe in the predictions and in the newly added Figure 13.
> >
> > In the future, we’d be interested in experiments involving pre-training with multiple tools and how this would affect learning efficiency when presented with tasks requiring novel tools.

---

### Author Response · Authors · 2025-11-19

We thank all the reviewers for the valuable feedback and for helping us improve the quality of our paper. We have updated the document to reflect your suggestions, adding clarification on the topics here discussed.

We have also corrected a typo in the x-axis labels of Figures 5 and 11, where all plots were previously labelled with 1M at the end. In our, we work selected the number of training steps based on the convergence of the algorithms in each task, meaning that not all tasks finish at 1M.
We note that this was a pure labelling typo and that all curves remain the same.

---

### Author Response · Authors · 2025-12-01
**Summary of changes after rebuttal for new AC**

After the news about the identity leak and re-assignment of papers to new area chairs, we provide here a summary of the discussion and changes to the paper that ensued.

Following the reviewers’ questions, we took the oportunity to incorporate their comments in the paper, helping us further strengthen the conclusions of our work with new baselines, visualizations, analyses and further discussion on most of the points raised by the reviewers. Below we highlight some of the main additions:

- **Per-task performance**: we add more discussion in the results section and plots with mean results over all DMC tasks (in figures 6 and 12). We note that in the benchmarks in question, it is common for performance to vary accross tasks, as observed in leading works in the field such as Dreamer-v3 and TD-MPC2.
- **Additional baseline**: per request of Reviewer eUBc we added a self-supervised baseline (CURL) to all the results. The addition of this baseline strenghtens the conclusions of our work, by showing that our approach significantly improves on alternative self-supervised approaches in terms of performance and sample-efficiency.
- **Standard deviation in Figure 1**: we add STD bars to Figure 1 after the request of reviewers eUBc and hc2R. This wasn’t added in the first draft as taking it across tasks with different difficulties, that converge to different rewards could result in a misleadingly high STD. Other works in the field (such as TD-MPC2 and Dreamer-v3) also do not report STD in aggregate metrics. Nevertheless, the new plots show that our approach does not increase the underlying standard deviation of existing methods, while showing strong performance in the upper and lower bounds vs. the baselines.

The items above were seen by Reviewer eUBc as the main weaknesses of the initial draft.

- **Clarity of Figure 4**: following the comments of reviewers eUBc and Y4MG we:

    **1)** **Re-did the figure** with clearer sequences, with predictions on a longer horizon allowing the movement to be more discernable and increased the size of the figure. We also improved the discussion about what this figure conveys.

    Here, we highlight that the visualization in Figure 4 is obtained with a model that was trained with a prediction horizon of 10 steps and used for prediction until time step 28, therefore resulting in some blurriness and divergence from the G.T.

    **2)** **Added a new visualization** on the Hammer task (App. A.5), better highlighting how the hammer is integrated in the learned agent features. Per request of reviewer eUBc we also provide a qualitative and quantitative analysis of the overlap between learned agent features and the supervisory agent masks used by alternative approaches. This analysis highlights how the learned features approximate the supervisory masks and adds support to the claims of the paper.

- **Clarification on Horizon ablation**: we agree with  Reviewer Y4MG that a gap of 500 in reward is significant and acknowledge that the initial draft was not fully clear in this point, when what we intended to convey is that we did not observe a **consistent** impact of the horizon lenght in the results, as DrQv2-EF improves vs. the baseline for every value that was tested.

    We made modifications to the text in the Ablation Study section in order to better get this point across.


The two items above were listed by Reviewer Y4MG as the two main weaknesses in the work (W1, W2, Q1, Q2 & Q3).

- **Analysis of computational overhead**: per request of the Reviewers eUBc and hc2R we added an analysis of the computational overhead of our approach in comparison to the baselines in Appendix A.4. The values provide transparency on the computational cost of our approach, which is compensated by the gains in sample-efficiency.

    We note that in real-world applications such as robotics, sample-efficiency can be considered of greater importance than computational efficiency, as the feasible number of trials becomes a much more limitative factor.

We again note that besides these changes, we added clarification in the paper regarding multiple other questions from the reviewers, which are detailed in the discussion below.

---

### Meta-Review · Area_Chair_n9Yy · 2026-01-06

**Summary:**

Review concerns from the reviewers that informed my suggested decision:

Reviewer eUBc
- "results on performance improvement does not seem to be consistent across the tasks"
- "why are other self-supervised image-based RL algorithms not included as comparison baselines"
- "DoorOpen task ... is not part of the MetaWorld tasks that require tool use ... you highlighted EF should be better in tasks that require tool use, why is the ablation study done on DoorOpen"
- "what Fig 4 is implying?"
- alternative explanation of ablation study
- "what is the failure mode? Can you characterize when EF is expected to help?"
- "can you quantify how much overlap there is between the learned agent feature with the supervised agent masks if available in some tasks?"

Reviewer Nv8q
- "The paper uses recurrent neural networks, while the overall efficacy of transformers in sequence generation is well-proven in the domains of natural language and vision."
- "Episode partitioning uses a fixed constant H for defining the future window. This choice could have been made more dynamic to remove any bias due to burden of predicting next H frames every time."
- "Are these results also applicable in non-robotics environments like standard control tasks (e.g. CartPole) in RL?"

Reviewer Y4MG
- distinction between the parts of the environment vs. parts of the robot's body
- The authors state that “horizon length does not significantly impact the results”. However, the “Prediction Horizon” graph in Figure 8 ..."

Reviewer hc2R
- "The authors propose that EF's improvements stem from (i) disentanglement of agent information allowing the RL algorithm to focus on agent control in early training stages and environmental interactions later, and (ii) regularization through imposing predictability in robot movements. However, there seems to be not enough evidence supporting this mechanism, particularly the claim about distinct training phases."

**Reviewer Concerns:**

The concerns of the reviews with lowest recommended scores appear to be addressed, most were about interpretations and explanations, rather than structural flaws. The authors included an additional requested self-supervised baseline.

**Reviewer Scores:**

- Reviewer eUBc: 4 -> 6
- Reviewer Nv8q: 8 -> 8
- Reviewer Y4MG: 4 -> 6
- Reviewer hc2R: 6 -> 6 (mentioned explicitly: https://openreview.net/forum?id=6itufi98Q3&noteId=3U9YQd7jaC)

---

### Decision · Program_Chairs · 2026-01-26

Accept (Poster)